# The Antioxidant, Anti-Apoptotic, and Proliferative Potency of Argan Oil against Betamethasone-Induced Oxidative Renal Damage in Rats

**DOI:** 10.3390/biology9110352

**Published:** 2020-10-23

**Authors:** Sahar Hassan Orabi, Tamer S. Allam, Sherif Mohamed Shawky, Enas Abd El-aziz Tahoun, Hanem K. Khalifa, Rafa Almeer, Mohamed M. Abdel-Daim, Nermeen Borai El-Borai, Ahmed Abdelmoniem Mousa

**Affiliations:** 1Department of Biochemistry and Chemistry of Nutrition, Faculty of Veterinary Medicine, University of Sadat City, Sadat City, Menofia 32897, Egypt; hanemkamalbasuni@yahoo.com (H.K.K.); ahmed.mousa@vet.usc.edu.eg (A.A.M.); 2Department of Clinical Pathology, Faculty of Veterinary Medicine, University of Sadat City, Sadat City, Menofia 32897, Egypt; tamerallam98@yahoo.com or; 3Department of Physiology, Faculty of Veterinary Medicine, University of Sadat City, Sadat City, Menofia 32897, Egypt; shsh00076@vet.usc.edu.eg; 4Department of Pathology, Faculty of Veterinary Medicine, University of Sadat City, Sadat City, Menofia 32897, Egypt; enastahoon35@yahoo.com; 5Department of Zoology, College of Science, King Saud University, P.O. Box 2455, Riyadh 11451, Saudi Arabia; ralmeer@ksu.edu.sa; 6Department of Pharmacology, Faculty of Veterinary Medicine, Suez Canal University, Ismailia 41522, Egypt; 7Department of Forensic Medicine & Toxicology, Faculty of Veterinary Medicine, University of Sadat City, Sadat City, Menofia 32897, Egypt

**Keywords:** betamethasone, argan oil, antioxidant, apoptosis, bax, Bcl-2, caspase-3, PCNA

## Abstract

**Simple Summary:**

The present study aimed to investigate the protective effect of argan oil against nephrotoxic effect following overdose and long-term administration of betamethasone. The results revealed that betamethasone induced hematological changes, including reduction of red blood cells with leukocytosis, neutrophilia, monocytosis, lymphocytopenia, and marked thrombocytopenia. Moreover, betamethasone caused significant increase of serum urea and creatinine levels; renal malondialdehyde and nitric oxide contents associated with significant decrease of reduced glutathione content. Betamethasone also caused vascular, degenerative, and inflammatory histopathological alterations in kidney tissue along with increase of Bax and caspase-3 expressions and decrease of B-cell lymphoma-2 (Bcl-2) and proliferating cell nuclear antigen (PCNA) expressions. Conversely, the concomitant administration of argan oil (0.5, 1 mL/kg) with betamethasone ameliorated the aforementioned hematological, biochemical, pathological, and histochemical adverse effects. In conclusion, overdose and long-term administration of betamethasone could induce hematological changes and severe renal damage mediated by oxidative, apoptotic and proliferative mechanisms via increasing renal functions biomarkers and altering oxidant/antioxidant status along with pathological lesions and imbalance of Bax/Bcl-2 ratio that positively correlates with up-regulation of caspase-3 and down-regulation of PCNA in kidney tissue. However, argan oil could potentially protect against betamethasone- induced renal damage, in a dose-dependent manner, via its antioxidant, anti-apoptotic and proliferative properties.

**Abstract:**

The present study aimed to investigate the protective effect of argan oil (AO) against nephrotoxic effects following overdose and long-term administration of betamethasone (BM). The phytochemical compositions of AO were assessed using GC/MS. Forty eight male Wister albino rats were divided into six groups and treated for 3 successive weeks. The control group was orally administrated distilled water daily, the BM group received BM (1 mg/kg, IM, day after day), AO/0.5 and AO/1 groups received AO (0.5 mL/kg, 1 mL/kg, orally, daily, respectively), BM + AO/0.5 group and BM + AO/1 group. The results revealed that BM induced hematological changes, including reduction of red blood cells with leukocytosis, neutrophilia, monocytosis, lymphocytopenia, and thrombocytopenia. Moreover, BM caused a significant increase of serum urea and creatinine levels, and renal malondialdehyde and nitric oxide contents with significant decrease of reduced glutathione content. BM also caused vascular, degenerative, and inflammatory histopathological alterations in kidney, along with an increase in the Bax/Bcl-2 ratio, activation of caspase-3, and decrease of proliferating cell nuclear antigen expression. Conversely, the concomitant administration of AO (0.5, 1 mL/kg) with BM ameliorated the aforementioned hematological, biochemical, pathological, and histochemical BM adverse effects. In conclusion, AO has protective effects against BM-induced renal damage, possibly via its antioxidant, anti-apoptotic, and proliferative properties.

## 1. Introduction

Glucocorticoids (GCs), a major class of adrenal steroid hormones, are the most important mediators of systemic inflammation and play an essential role in the maintenance of a variety of metabolic and homeostatic functions [1,2]. Synthetic glucocorticoids, including betamethasone (BM), are widely used as pharmacological agents used to substitute the deficiency or the complete absence of the endogenous hormones and have anti-inflammatory and immunosuppressive properties [3,4].

The clinical and therapeutic benefits of synthetic glucocorticoids are associated with many adverse side effects, particularly during high-dose and/or long-term administration [5]. Among these adverse effects, neuropsychiatric disorders, hyperglycemia, hypertension, obesity, gastritis, osteoporosis, glaucoma, cataracts, and cardiovascular disease were recorded following exposure to synthetic glucocorticoids [6,7,8,9,10].

Mounting evidence has demonstrated the implication of oxidative stress in the mechanisms of various GC-induced metabolic disorders, as well as adverse effects on various tissues, including liver, kidney, brain, and heart, in addition to bones [11,12,13].Thus, natural products rich in antioxidants are needed for curbing such adverse effects, mainly via its remarkable ability to scavenge ROS, modulate antioxidant/pro-oxidant enzymes and/or transcription factors, and enhance cellular antioxidant enzymes and formation of bioactive metabolites [14].

Argan oil (AO) is natural vegetable edible oil extracted from seeds and fruits of the argan tree (*Argania spinosa* L.), a tree endemic to Morocco [15]. AO is recommended for nutritional and pharmacological purposes, mainly due to its various chemical constituents, including saturated and unsaturated fatty acids, saponins, sterols, squalene, tocopherols, minerals, phenolic compounds, coenzyme Q10, and melatonin [16,17]. Moreover, AO is widely used as an ingredient in different cosmetic products such as shampoos and moisturizers [18]. Traditionally, AO is effective as a topical application for the healing of burns and for the treatment of several skin diseases [16].

Interestingly, AO has therapeutic value in cardiovascular disorders associated with obesity, hypertension, and insulin resistance [16]. Moreover, previous literature confirmed the antioxidant, anti-inflammatory, anti-mutagenic, and anti-carcinogenic activities of AO [19,20], as well as its neuro- and hepatorenal protective effects [19,21,22].

In this context, the current study was designed firstly to investigate the effect of overdose and long-term administration of betamethasone on kidney of rats and the prospective ameliorative effect of argan oil, and also to clarify the mechanistic role of oxidative stress, highlighting the relevance between the Bax/Bcl-2 ratio, caspase-3 activity, and apoptosis, as well as renal proliferation.

## 2. Materials and Methods

### 2.1. Chemicals 

Betamethasone, BM (Diprofosinjectable^®^, Schering-Plough Company, Kenilworth, New Jersey, NJ, USA) was obtained from a local pharmacy in Sadat City, Egypt. Argan oil (AO) was purchased from a Moroccan market in Montpellier, France. Diagnostic kits for assaying serum and tissue biochemical parameters were purchased from Biodiagnostic Company, Dokki, Giza, Egypt. Other utilized chemicals and reagents were of analytical grade and commercially available.

### 2.2. Gas Chromatography–Mass Spectrometry (GC–MS) Analysis of Argan Oil

The argan oil phytochemical analysis was conducted by using a Trace GC1300-TSQ mass spectrometer (Thermo Scientific, Austin, TX, USA) with a direct capillary column TG–5MS (30 m × 0.25 mm × 0.25 µm film thickness). The column oven temperature was firstly kept at 50 °C and increased by 5 °C/min to 250 °C and held at that temperature for 2 min; finally, the temperature was raised by 25 °C/min to reach 300 °C and held at that temperature for 2 min. The temperatures of the injector and MS transfer line were maintained at 250 and 260 °C, respectively. Helium was used at a steady flow rate of 1 mL/min as the carrier gas. The solvent delay was 3 min, and 1 μL each of the diluted samples were automatically injected using an AS1300 autosampler coupled with GC in the split mode (Thermo Scientific, Austin, TX, USA). EI mass spectra were collected at ionizing voltages of 70 eV over the range of m/z 50–650 in full scan mode. The temperature of the ion source was set at 250 °C. Comparing their retention times and mass spectra with those of the mass spectral database WILEY 09 and NIST 11 (New York City, NY, USA), the components were identified.

### 2.3. Animals and Experimental Protocol

Forty-eight 6 week old adult male Wistar albino rats (120–130 g) were purchased from Al-Zyade Experimental Animals Production Center, Giza, Egypt. Rats were housed in plastic cages with stainless-steel grid tops and kept at a well-ventilated laboratory animal room (23 ± 3 °C, 45–55% relative humidity, natural daily dark/light cycle). Rats received a balanced diet and tap water ad libitum for one week before starting the experiment. This study was ethically approved by the International Animal Care and Use Committee (IACUC), Faculty of Veterinary Medicine, University of Sadat City, Egypt (Approval No.VUSC-012-2-19), which follows the Guide for the Care and Use of Laboratory Animals 8th edition. Washington (DC): National Academies Press (US); 2011.

Rats were randomly divided into six equal groups (8 rats each) as follows:

Control group: Rats were orally administrated distilled water, daily for 3 successive weeks.

Betamethasone (BM) group: Rats were intramuscularly injected BM at a dose of 1 mg/kg [23], day after day for 3 successive weeks.

Argan oil 0.5 mL (AO/0.5) group: Rats were orally administrated AO as an experimental trial dose of 0.5 mL/kg, daily for 3 successive weeks.

Argan oil 1 mL (AO/1) group: Rats were administrated orally AO as an experimental trial dose of 1 mL/kg, daily for 3 successive weeks.

Betamethasone and Argan oil 0.5 mL (BM+AO/0.5) group: Rats were intramuscularly injected BM (1 mg/kg) and then orally administrated AO (0.5 mL/kg).

Betamethasone and Argan oil 1 mL (BM+AO/1) group: Rats were intramuscularly injected BM (1 mg/kg) and orally administrated AO (1 mL/kg).

### 2.4. Sample Collection and Preparation

By the end of the experiment, all rats were fasted, and blood samples were collected from the retro-orbital venous plexus (under inhalation anesthesia of isoflurane) on EDTA for hematological assays, while another blood sample was collected and centrifuged at 3000 rpm for serum separation and then kept at −20 °C for further serum biochemical analyses. After sacrificing of rats by cervical dislocation, the kidneys from each rat were excised immediately; one kidney was kept at −80 °C for further tissue biochemical analysis, while the other kidney was fixed in 10% neutral buffered formalin for histopathology and immunohistochemistry examinations.

The preserved kidney from each rat was washed in normal saline solution and blotted over filter paper and then perfused in a cold phosphate-buffered saline (pH 7.4) containing 0.16 mg/mL heparin to remove blood clots. Then 1 g of kidney tissue was homogenized in 9 volumes cold PBS (pH 7.4) using an automatic tissue homogenizer. The tissue homogenate was centrifuged at 4 °C and 4000 rpm for 15 min, and the supernatant was collected and kept at –80 °C for the measuring of malondialdehyde (MDA), nitric oxide (NO), and reduced glutathione (GSH) contents following the kits’ instructions.

### 2.5. Investigation of Hematological Parameters

Hematological changes were investigated through the evaluation of erythrogram, leukogram, and blood platelets. The erythrogram parameters included red blood cell count (RBCs), hemoglobin concentration (Hb), packed cell volume (PCV), mean corpuscular volume (MCV), mean corpuscular hemoglobin (MCH), and mean corpuscular hemoglobin concentration (MCHC), while the leukogram parameters included total white blood cell count (WBCs) and differential leukocytic counts. All the above-mentioned parameters were assessed by a BeneSphera™ Brand3 Part differential veterinary hematology analyzer H23 (Avantor Performance Materials, Inc, The Netherlands, Holanda, Model: H23vet. Serial No: 931716004) and according to Grindem [24].

### 2.6. Assessment of Serum Renal Function Biomarkers 

Serum urea and creatinine levels were estimated using commercial kits purchased from Biodiagnostic Company (Dokki, Giza, Egypt). Urea (CAT.NO.UR 2110) and creatinine (CAT. NO.CR 1250) were performed according to the manufacturer’s instructions.

### 2.7. Assessment of Renal Oxidant/Antioxidant Biomarkers

Oxidant/antioxidant biomarkers, including MDA, NO, and GSH contents were assessed in kidney tissue homogenate using commercial kits of MDA (CAT.NO.MD 2529), NO (CAT.NO. 2533), and GSH (CAT.NO. GR 2511) that were purchased from Biodiagnostic Company.

### 2.8. Histopathological Examination

The fixed kidney specimens were trimmed and processed for paraffin sections (4 μm thick) using a microtome (LEICA RM 2135) then routinely stained with hematoxylin and eosin stain (H&E) according to Bancroft and Layton [25]. Histopathological examination and photographing were done using a digital Leica photomicroscope (LEICA DMLB Germany). All sections of kidney samples were examined for glomerular, tubular, and interstitial alterations. Semi-quantitation for each of the parameters, namely vascular (congestion, edema), degenerative, and inflammatory changes were evaluated by scoring the degree of the severity following the scoring system described by Oyouni et al. [26] with some modifications as follows: (−) normal appearance and absence of pathological lesion 0%, (+) mild (<25% of sections), (++) moderate (26–50% of sections), (+++) severe (51–75% of sections), and (++++) very severe (>75% of sections).

### 2.9. Immunohistochemical Investigation

The immune-staining method for localization of Bax (Bcl-2-associated X), B-cell lymphoma-2 (Bcl-2), and caspase-3 was performed following the method adopted by Hassanein et al. [27] and following the method adopted by Zhu et al. [28] for localization of proliferating cell nuclear antigen (PCNA). The formalin-fixed kidney sections were deparaffinized, rehydrated in alcohol solutions, incubated in 3% H_2_O_2_, and then incubated with anti-Bax (diluted 1:100, Santa Cruz Biotechnology Santa Cruz, CA, USA), anti-Bcl-2 (diluted 1:200, Santa Cruz Biotechnology, Santa Cruz, CA, USA), anti-caspase-3 (diluted 1:200, Abcam, Cambridge, MA, USA), or anti-PCNA (diluted 1:300; Abcam, Cambridge, MA, USA) antibodies overnight at 4 °C as primary antibodies. After rinsing with phosphate buffered saline, they were incubated with the appropriate biotinylated secondary antibodies according to the Vecta stain Elite ABC Kit (Vector Laboratories, Burlingame, CA, USA) for 30 min at room temperature. The immune reactions were visualized by using diaminobenzidene (DAB, Sigma Chemical Co, St. Louis, Missouri, USA) under light microscopy and then counterstained with Mayer’s hematoxylin (Sigma-Aldrich, St. Louis, MO, USA) followed by routine dehydration in alcohol, clearing in xylene, and mounting using the Aquatex fluid (Merk KGaA, Darmstadt, Germany) under a cover slip. Semi-quantification analysis of apoptotic index (AI) and PCNA expression represented by the percentage of positively immune stained cells (brown cytoplasm for Bax, Bcl-2 and brown nucleus for caspase-3, PCNA) by counting at least 1000 cells per slide, subdivided into 10 random fields. AI% = (number of positive cells/total number of calculated cells) × 100, HPF (40×), as previously described by Rahman et al. [29], and then the ratio between Bax and Bcl-2 was calculated.

### 2.10. Statistical Analysis 

Statistical analyses were carried out by one-way ANOVA followed by Duncan’s multiple range test for post hoc analysis. All statistical analyses were performed using SPSS (Statistical Package for Social Sciences) Version 16 released in 2007, SPSS Inc., Chicago, IL, USA. Values were presented as mean ± SE. Differences were considered statistically significant at *p* < 0.05.

## 3. Results

### 3.1. The Phytochemical Components of Argan Oil

The results of the phytochemical analysis of AO using GC–MS can be seen in Table 1, which shows the presence of fifteen phyto-constituents, with cyclopentasiloxane, decamethyl-, and phenol, 2,6-bis(1,1-dimethylethyl)-4-methyl being the main constituents.

### 3.2. Argan Oil Ameliorated Betamethasone-Induced Changes in Erythrogram Parameters of Rats

The effect of administration of BM and/or AO on erythrogram parameters is represented in Table 2; rats exposed to overdose and long-term administration of BM showed significant (*p* < 0.05) reduction in the values of RBCs, Hb, PCV, and MCHC with insignificant changes in MCV and MCH values, when compared with control rats. In contrast, compared with the BM group, rats of the BM+AO/0.5 group revealed a significant increase only in the value of Hb, while those of BM+AO/1 showed a significant increase in the values of Hb and PCV with normalization of the other erythrogram values in both groups. No significant changes were recorded either in AO/0.5-and AO/1-treated rats, compared to those of control rats.

### 3.3. Argan Oil Alleviated Betamethasone-Induced Changes in Leukogram and Blood Platelets of Rats

Referring to changes of leukogram parameters, data in Table 3 demonstrated that long-term and overdose administration of BM induced significant (*p* < 0.05) increase in counts of TWBCs, neutrophils, and monocytes with significant decreases in counts of lymphocytes and platelets, when compared to those of the control group. Rats administrated AO at a dose of either 0.5 or 1 mL concomitantly with BM showed improvement of the corresponding values of the BM group, which was significant in the counts of TWBCs, neutrophils, monocytes, and platelets; however, it was insignificant in lymphocytic count. Interestingly, concomitant administration of AO at both doses kept the normal control values of TWBCs and neutrophils. Comparing with the control group, administration of AO/0.5 or AO/1 resulted in insignificant variations in platelet count and leukogram parameters, except a significant increase in monocytes count.

### 3.4. Argan Oil Reduced Betamethasone-Induced Elevation in Serum Kidney Functions Biomarkers of Rats

There was significant (*p* < 0.05) elevation in values of serum urea and creatinine levels in BM-injected rats, compared to those of the control group. However, rats administrated AO at a dose of either 0.5 or 1 mL concomitantly with BM, showed a significant reduction in these values compared to those of the BM group. Meanwhile, AO at a dose of 1 mL kept all values within normal control levels. Argan alone at any dose showed no significant effects on serum levels of urea and creatinine, when compared to those of the control group (Figure 1).

### 3.5. Argan Oil Improved Betamethasone-Induced Alterations in Renal Oxidant/Antioxidant Biomarkers of Rats

The effects of BM and/or AO on the renal oxidant/antioxidant biomarkers of rats are shown in Figure 2. Significant elevation of MDA and NO contents along with significant (*p* < 0.05) reduction of GSH content were recorded in renal tissue homogenates of BM-exposed rats, compared to the control group. Conversely, co-administration of BM and AO (0.5 or 1 mL/kg) showed improvement represented by significant reduction in renal MDA and NO levels with an increase of GSH content, compared with the BM group. On the other hand, AO alone at both doses had no significant effects on the renal oxidant/ antioxidant status, compared to those in the control group.

### 3.6. Argan Oil Ameliorated Betamethasone-Induced Alterations in Renal Histoarchitecture of Rats

Table 4 and Figure 3 illustrated the renal histopathological alterations. Kidneys of the control group showed normal architecture of kidney glomeruli and renal tubules with intact well-organized cellular boundaries (Figure 3A). In contrast, kidney of the BM group showed marked vascular changes, including congestion, and perivascular and interstitial edema associated with degenerative changes as proved by sever atrophy of some proximal convoluted renal tubules with sever nuclear pyknosis, and dilatation of some renal tubules with hyaline and cellular casts. Inflammatory cell infiltration was also seen in renal tissue of the BM group (Figure 3B). Similar to the control group, kidneys of AO/0.5 (Figure 3C) and AO/1 (Figure 3D) groups showed normal architecture of glomeruli and renal tubules. However, administration of AO (0.5 or 1 mL/kg) to BM-treated rats resulted in improvement in renal tissue architecture, except for mild congestion, mild edema, mild nuclear pyknosis of some renal tubular cells, and few inflammatory cell infiltrations in the interstitial tissue (Figure 3E, 3F, respectively).

### 3.7. Argan Oil Modulated Betamethasone-Induced Increase of Bax and Caspase-3 and Decrease of Bcl-2 and PCNA Expressions in Renal Tissues of Rats

Table 5 and Figure 4, Figure 5, Figure 6 and Figure 7 present the immune reactivity for Bax, Bcl-2, caspase-3, and PCNA, respectively, in renal tissues of different treated groups. In addition, the calculated ratio of Bax to Bcl-2 is presented in Table 5, which showed the highest Bax/Bcl-2 ratio in the BM group with a significant increase in the ratio, namely 5.27 ± 0.73, compared to the control group (0.36 ± 0.06). Conversely, a significant decrease was recorded in the Bax/Bcl-2 ratio of BM+AO/0.5 and BM+AO/1 groups, being 0.56 ± 0.03 and 0.49 ± 0.02, respectively, when compared to BM group. Nevertheless, no significant variations were recorded in the Bax/Bcl-2 ratio either between AO/0.5 or AO/1 groups and control group.

Regarding the results of Bax immune reactivity of renal tubular cells, control (Figure 4A), AO/0.5 (Figure 4C), and AO/1 (Figure 4D) groups showed slight cytoplasmic expression of Bax immune-staining in the renal tubular cells with percentage of positive cells being 9.85 ± 0.61, 8.37 ± 0.54, and 7.25 ± 0.31, respectively. In contrary, the BM group showed marked cytoplasmic expression of Bax immune-staining in most renal tubular cells of both proximal and distal convoluted tubules, with percentage of positive cells being 28.24 ± 1.43. A significant decrease in Bax immune-reactivity was observed in BM + AO/0.5 (Figure 4E) and BM + AO/1 (Figure 4F) groups, compared with the BM group, with percentage of positive cells being 15.94 ± 1.01 and 14.21 ± 1.05, respectively (Table 5).

Kidney sections of control group (Figure 5A) showed cytoplasmic immune reactivity of renal tubules for Bcl-2 with percentage of positive cells being 27.33 ± 0.88. In contrast, a marked decrease in the cytoplasmic Bcl-2 immune reactivity was recorded in renal tubular cells of the BM group (Figure 5B), with percentage of positive cells being 5.36 ± 0.17. Similar to the control group, cytoplasmic immune reactivity was observed in renal tubules of AO/0.5 (Figure 5C) and AO/1 (Figure 5D) groups with percentage of positive cells being 24.35 ± 0.54 and 25.40 ± 0.44, respectively. This immune reactivity was significantly increased in renal tubules of BM + AO/0.5 (Figure 5E) and BM+AO/1(Figure 5F) groups, with percentage of positive cells being 28.67 ± 0.50 and 28.82 ± 0.13, respectively, when compared to the BM group.

Concerning caspase-3 immune expression in the control and in the experimental groups is demonstrated in Figure 6. Caspase-3 immunopositivity was slight in the control (Figure 6A), AO/0.5 (Figure 6C), and AO/1 (Figure 6D) groups, respectively, with percentage of positive cells being 7.07 ± 0.72, 6.42 ± 0.36, and 5.49 ± 0.64, respectively. Moreover, significant increases in nuclear caspase-3 immune expression in glomerular and renal tubular cells were observed in kidney sections of the BM group (Figure 6B), with the percentage of positive cells being 39.87 ± 2.00 compared to the control group. It was observed that this intense specific immune-staining was significantly reduced in the BM+AO/0.5 (Figure 6E) and BM+AO/1(Figure 6F) groups, compared with the BM group, with percentage of positive cells being 18.48 ± 1.73 and 15.75 ± 1.92, respectively.

Referring to PCNA immune-expression within the renal tubular epithelium and interstitial inflammatory cells, control (Figure 7A), AO/0.5 (Figure 7C), and AO/1 (Figure 7D) groups showed nuclear immune-reactivity in the renal tubular epithelium, with percentage of positive cells being 32.75 ± 0.39, 33.35 ± 0.49, and 34.13 ± 0.46, respectively. Moreover, significant decreases in nuclear PCNA immune-reactivity in renal epithelium were observed in kidney sections of the BM group (Figure 7B), with percentage of positive cells being 10.67 ± 0.36 compared to the control group. A significant increase in PCNA immune-reactivity within the renal tubular epithelium and interstitial inflammatory cells was seen in both BM + AO/0.5 (Figure 7E) and BM + AO/1 (Figure 7F) groups, with percentage of positive cells being 30.48 ± 0.43 and 31.70 ± 0.31, respectively, compared with BM group (Table 5).

## 4. Discussion

Betamethasone is a synthetic glucocorticoid that has anti-inflammatory and immunosuppressive properties [3,4]. Despite its pharmacological benefits, overdose or long-term exposure to BM causes drastic and long-lasting side effects. Otherwise, argan oil has been demonstrated for its antioxidant, anti-inflammatory, and anti-apoptotic properties [19,20].

The current study revealed that BM caused anemia manifested by significant decreases in values of RBCs, Hb, and PCV and that it could be erythropoietin-resistant anemia due to oxidative stress [30]. Interestingly, the erythropoietin hormone is produced mainly by the kidney from jaxta glomerular cells and regulates the erythropoiesis process [31]. Hence, erythropoietin deficiency due to renal oxidative damage induced by BM might be the reason behind the observed normocytic normochromic anemia that evidenced by insignificant values of MCV and MCH. In addition, thrombopoietin hormone is also produced by the kidney and regulates the thrombopoiesis process [32], and this could explain the reported thrombocytopenia in BM-injected rats. 

Regarding the leukogram, this study revealed significant increases in total WBCs, neutrophils, and monocytes with significant decreases in lymphocytes. These results were in line with the previous findings of Cruz-Topet and Cidlowski [33], who reported that chronic exposure to BM has an immunosuppressive effect. A decline in blood lymphocytes and an increase in neutrophils were also observed following subacute BM administration [34].

The role of kidney in excretion of xenobiotics predisposes it to structural and/or functional damage [35]. Drugs are one of the kidney-affecting agents, and nephrotoxicity remains a substantial adverse drug reaction following administration of various pharmacological agents, including glucocorticoids [36].

The current findings revealed that BM induced renal damage, represented by a marked increment of serum urea and creatinine levels. In association with these findings, vascular, degenerative and inflammatory pathological changes were observed in renal tissue of BM-injected rats (Figure 3). These findings were in line with those of Sharma et al. [34], who recorded increases in serum urea and creatinine levels in a dose-dependent manner after subacute exposure to BM. In addition, Jain et al. [37] reported elevation in serum urea and creatinine levels in male rats after prenatal exposure to glucocorticoids and a decline in glomerular filtration. Therefore, the increment in kidney function biomarkers may be attributed to the renal damage induced by overdose and long-term administration of BM, resulting in reduction of glomerular filtration rate, impairment in renal tubular reabsorption, and decrease of urea and creatinine excretion, thus increasing their level in the blood [38]. Additionally, one of the major possible explanations of BM-induced renal damage is the initiation of oxidative stress. Over-generation of ROS ultimately elevates lipid peroxidation and reduces the cellular antioxidants in the renal tissue, as evidenced by the recorded elevation in the levels of MDA and NO and reduction of GSH contents (Figure 2). In accordance with our findings, Verhaeghe et al. [39] stated that BM administration caused an elevation in MDA and NO levels, as well as reduction of glutathione peroxidase activity. In addition, Sato et al. [40] reported that glucocorticoid induced oxidative stress by generating ROS in the rat brain and depletion in the activity of antioxidant enzymes.

As a consequence of renal oxidative damage induced by BM, our immunohistochemistry findings strongly suggested activation of the cell death pathway indicated by changes in the balance of the apoptosis regulatory proteins, Bax, Bcl-2, and caspase-3 in renal tissue following overdose and long-term exposure of BM. Currently, there is no report about the effects of BM on these regulatory proteins in kidney; however similar to our findings, Pedrana et al. [41] reported that in utero exposure to BM increased Bax expression and decreased Bcl-2 expression in lamb testicular tissue. Apoptosis is a complex biological process that regulates cell survival by removal of diseased or deteriorated cells [42]. Bax and Bcl-2 are pro-apoptotic and anti-apoptotic regulatory proteins, respectively that work as pairs to regulate the apoptosis occurrence and progression [43]. Simply, Bax/Bcl-2 ratio controls cell death or survival following any apoptotic stimulus. Bax promotes apoptosis by increasing the release of apoptogenic factors [44]; however, Bcl-2 prevents apoptotic cell death by reducing the release of apoptogenic factors such as cytochrome *c* and apoptosis-inducing factor from the intermembrane space of mitochondria [45]. Moreover, activation of caspase-3, an apoptotic marker, indicates irreversible cell death [46]. It is well known that oxidative stress is often responsible for the mitochondria-mediated signaling pathway of apoptosis [47,48], and generation of large amount of ROS decreases the expression of Bcl-2 [49], resulting in increased release of cytochrome *c* that promotes the activation of caspase-3 leading to apoptotic cell death [50]. These changes eventually resulted in an increased number of apoptotic cells in the kidney of BM-exposed rats, perhaps via the imbalance between pro- and anti-apoptotic proteins.

PCNA is a nuclear protein that has a substantial role in DNA synthesis, repair, and replication. It is an indicative marker of proliferating cells [51]. Our immunohistochemical findings demonstrated reduction in renal cell proliferation by suppressing PCNA expression, as indicated by decreases of nuclear PCNA immunoreactivity in renal epithelia. Up till now, no available data have been reported about the effect of BM on PCNA expression in renal tissue. However, Pedrana et al. [41] reported a decrease in PCNA expression in lamb testicular tissue following in utero exposure to BM. We, therefore, hypothesized that down-regulation of PCNA is mediated by oxidative renal damage induced by BM.

Concerning the GC/MS findings, the two most abundant compounds found in AO are cyclopentasiloxane, decamethyl-, and phenol, 2,6-bis(1,1-dimethylethyl)-4-methyl. Cyclopentasiloxane, decamethyl- is classified as a cyclomethicone that is commonly used in cosmetics, such as deodorants, sunblock, hair sprays, and skin care products. It is also used as part of silicone-based personal lubricants [52], while phenol, 2,6-bis(1,1-dimethylethyl)-4-methyl is the phenolic antioxidant also known as 2,6-ditertiary-butyl paracresol (DBPC) or butylatedhydroxytoluene (BHT) [53].

On the other hand, regarding the cytoprotective and ameliorative effects of AO against BM-hazards, our results revealed that argan oil normalized the altered hematological values in rats (Table 2 and Table 3). Our findings were parallel to those of Şekeroğlu et al. [20], who recorded that AO administration improved the alterations of hematological parameters induced by acrylamide toxicity. The improved hematological parameters may be attributed to the recorded improvement in kidney functions following AO treatment due to the essential role of kidney in the production of the hematopoietic growth factors erythropoietin and thrombopoietin hormones, which regulate the erythropoiesis and thrombopoiesis processes [54]. Moreover, AO could ameliorate the hematological alterations induced by BM, mainly through its antioxidant and anti-inflammatory activities [19].

Moreover, AO exerted nephroprotective effects manifested by the reduction of the elevated urea and creatinine levels and improvement of the renal architecture. Consistent with previous studies, AO improved the kidney function biomarkers following exposure to hydrogen peroxide [55], lead [56], and Na-fluoride [22] toxicity. These protective effects of AO against BM-induced nephrotoxicity may be closely related to its powerful antioxidant effect evidenced by the decrease of MDA and NO levels and the increase of GSH content in renal tissue recorded in this study and previous studies [20,22,57]. The antioxidant properties are obviously due to its content of bioactive molecules and antioxidants such as vanillic acid caffeic acid, ferulic acid, resorcinol, catechin, phenolic compounds, coenzyme Q10, and melatonin [16,17,55].

Argan oil ameliorated the hazard effects of BM by reducing the Bax/Bcl-2 ratio that is closely-linked to the down-regulation of caspase-3 in renal tissue, which could be explained by the anti-oxidative properties of argan oil [20]. These findings suggest that Bax, Bcl-2, and caspase-3 play crucial roles in the repair process of damaged renal tubules [58]. Additionally, AO improved the proliferation and regeneration of kidney, as a compensatory mechanism to replace the degenerative changes via up-regulation of nuclear PCNA of the renal tubular epithelium. The regenerative properties of AO may be due to its content of β-carotene, which is essential for the repair and regeneration of damaged cells [59]. This is consistent with the idea that reduction of the Bax/Bcl-2 ratio following AO co-administration could reduce apoptosis and promote cell survival by minimizing mitochondrial membrane permeability and down-regulating caspases, as well as enhancing cell proliferation [60]. Consequently, argan oil ameliorated BM-induced hematological changes and biochemical, pathological, and histochemical alterations in kidneys of rats.

## 5. Conclusions

The aforementioned findings revealed that overdose and long-term administration of betamethasone could induce hematological changes and severe renal damage mediated by oxidative, apoptotic, and proliferative mechanisms via increasing renal function biomarkers and altering oxidant/antioxidant status along with pathological lesions and imbalance of the Bax/Bcl-2 ratio that positively correlates with up-regulation of caspase-3 and down-regulation of PCNA in kidney tissue. However, argan oil could potentially protect against BM-induced renal damage, in a dose-dependent manner, via its antioxidant, anti-apoptotic, and proliferative properties.

## Figures and Tables

**Figure 1 biology-09-00352-f001:**
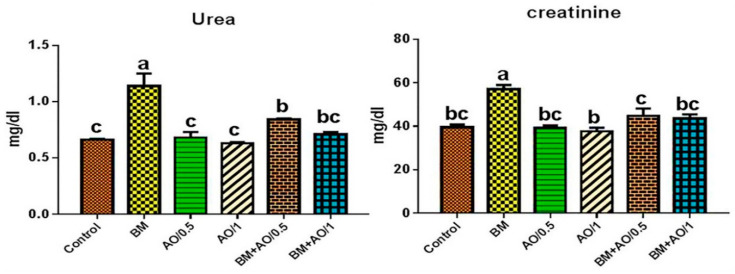
Effect of betamethasone and/or argan oil on serum renal function biomarkers of different treated groups (*n* = 8). Different superscripts (a,b,c) indicate significant differences at *p* < 0.05. BM: betamethasone, AO: argan oil.

**Figure 2 biology-09-00352-f002:**
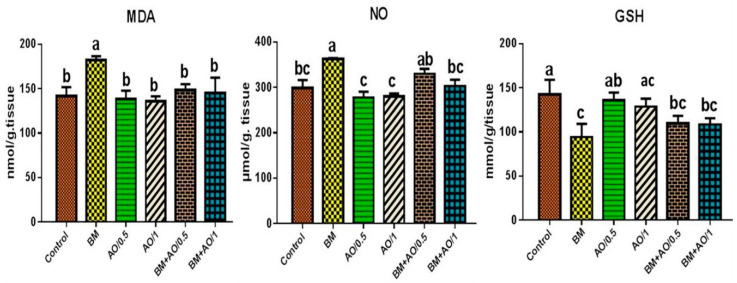
Effect of betamethasone and/or argan oil on renal oxidant/antioxidant biomarkers different treated groups (*n* = 8). Different letters (a,b,c) indicate significant differences at *p* < 0.05. BM: betamethasone, AO: argan oil, MDA: malondialdehyde, NO: nitric oxide, GSH: reduced glutathione.

**Figure 3 biology-09-00352-f003:**
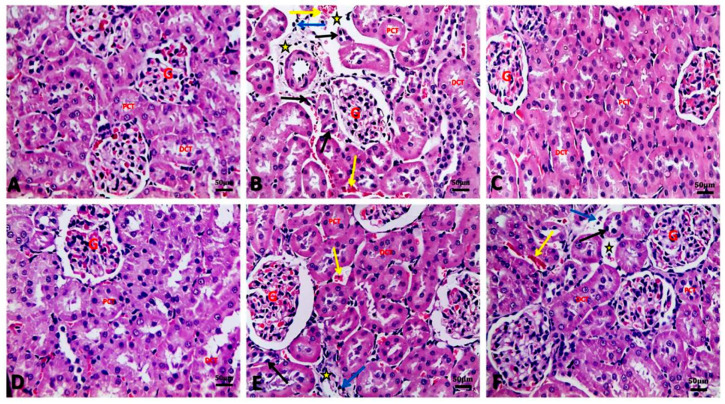
Representative photomicrographs of histopathological alterations in kidney sections of different groups (H&E stain X_20_; scale bar 50 μm; star: edema, yellow arrow: congestion, black arrow: tubular atrophy with nuclear pyknosis, blue arrow: inflammatory cell infiltration, G: glomeruli, PCT: proximal convoluted tubules, DCT: distal convoluted tubules). (**A**) Control group showing normal architecture of kidney glomeruli and renal tubules (proximal and distal convoluted tubules) with intact well organized cellular boundary. (**B**) BM group showing perivascular edema and congestion in the interstitial tissue in between renal tubules, atrophied proximal renal tubules with nuclear pyknosis, and inflammatory cell infiltration. (**C**) AO/0.5 and (**D**) AO/1 showing normal architecture of glomeruli and renal tubules as control group. (**E**) BM+AO/0.5 group and (**F**) BM+AO/1 group showing mild congestion, mild edema, and mild nuclear pyknosis of some renal tubular cells and mild inflammatory cell infiltration in the interstitial tissue.

**Figure 4 biology-09-00352-f004:**
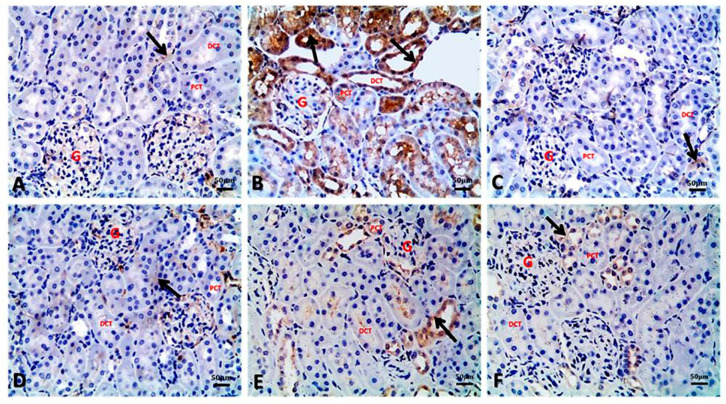
Representative photomicrographs of immuno-histochemical localization of Bax in kidney sections of different groups (Bax IHC, bar = 50 µm). (**A**) Control group showing slight cytoplasmic expression of Bax immune-staining in the renal tubular cells. (**B**) BM group showing marked cytoplasmic immune-staining of Bax in most renal tubular cells of both proximal and distal convoluted tubules. (**C**) AO/0.5 and (**D**) AO/1 showing slight cytoplasmic immune-staining of Bax as control group. (**E**) BM + AO/0.5 showing mild cytoplasmic immune-staining of Bax within some renal tubules. (**F**) BM + AO/1 showing mild cytoplasmic immune-staining of Bax within the renal tubular cells.

**Figure 5 biology-09-00352-f005:**
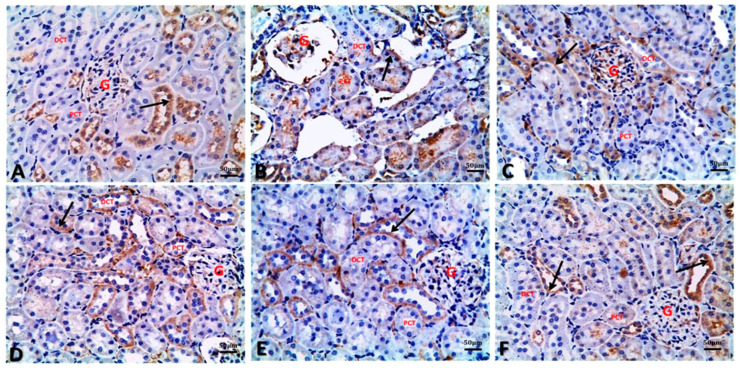
Representative photomicrographs of immuno-histochemical localization of Bcl-2 in kidney sections of different groups (Bcl-2 IHC, bar = 50 µm). (**A**) Control group showing cytoplasmic Bcl-2 immune-staining in the renal tubular epithelium. (**B**) BM group showing marked decrease in Bcl-2 immune-expressions in the renal tubular epithelium. (**C**) AO/0.5 and (**D**) AO/1 showing cytoplasmic Bcl-2 immune-staining as control group. (**E**) BM+AO/0.5 showing marked increase in the cytoplasmic immune-staining of Bcl-2 within the renal tubules. (**F**): BM+AO/1 showing marked increase in the cytoplasmic Bcl-2 immune-staining within the renal tubular epithelium.

**Figure 6 biology-09-00352-f006:**
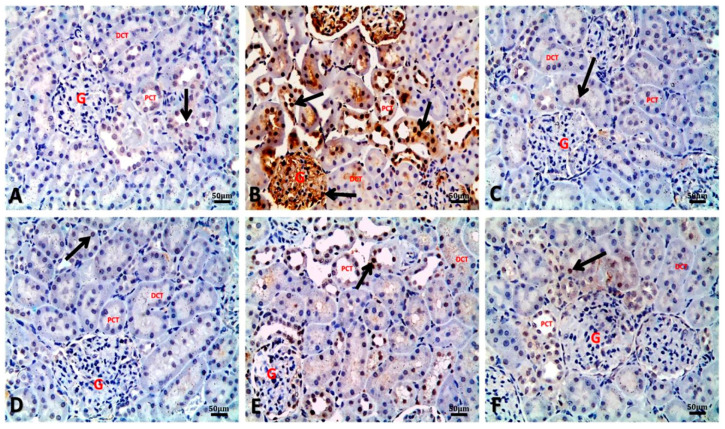
Representative photomicrographs of immuno-histochemical localization of caspase-3 in kidney sections of different groups (caspase-3 IHC, bar = 50 µm). (**A**) Control group showing slight nuclear immune-staining of caspase-3 in the renal tubular cells. (**B**) BM group showing marked dark immune-staining of caspase-3 in both glomerular and renal tubular cells. (**C**) AO/0.5 and (**D**) AO/1 showing slight nuclear immune-staining of caspase-3 as control group. (**E**) BM + AO/0.5 showing mild immune-staining of caspase-3 within some renal tubules. (**F**) BM + AO/1 showing mild immune-staining of caspase-3 within the renal tubular cells.

**Figure 7 biology-09-00352-f007:**
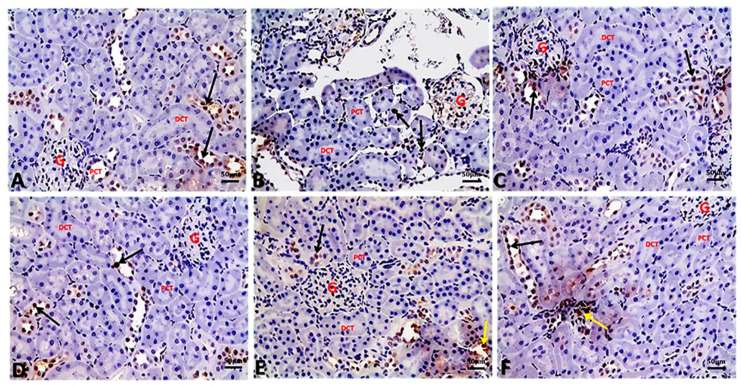
Representative photomicrographs of immuno-histochemical localization of PCNA in kidney sections of different groups (PCNA IHC, bar = 50 µm, black arrow: nuclear immune-expression of renal epithelium, yellow arrow: nuclear immune-staining within interstitial inflammatory cells). (**A**) Control group showing nuclear immune-staining of PCNA within the epithelial lining of the renal tubules. (**B**) BM group showing marked decrease in PCNA immuno-staining within the renal tubular epithelium. (**C**) AO/0.5 and (**D**) AO/1 showing nuclear immune-staining of PCNA within the renal tubules as control group. (**E**) BM + AO/0.5 showing nuclear immune-staining of PCNA within the renal tubules and interstitial inflammatory cells in-between renal tubules. (**F**) BM + AO/1 showing nuclear immune-staining of PCNA within the renal tubules and interstitial inflammatory cells.

**Table 1 biology-09-00352-t001:** The phytochemical compositions of argan oil.

Compound Name	RT/min	Area %	MW	MF
Cyclopentasiloxane, decamethyl-	5.05	28.3	370	C_10_H_30_O_5_Si_5_
Decyloxyamine	6.9	0.49	168	C_10_H_23_NO
Nonadecane	7.07	0.45	268	C_19_H_40_
Hexadecane	8.62	0.92	226	C_16_H_34_
Phytol	8.76	0.52	296	C_20_H_40_O
2-Piperidinone, N-[4-bromo-n-butyl]-	8.85	0.5	233	C_9_H_16_BrNO
Tridecane, 3-methyl-	8.97	0.41	198	C_14_H_30_
7-Hexadecenal, (Z)-	9.67	0.79	238	C_16_H_30_O
8-Hexadecenal, 14-methyl-, (Z)-	9.78	0.54	252	C_17_H_32_O
Phenol, 2,6-BIS(1,1-Dimethylethyl)-4 –Methyl	11.55	13.13	220	C_15_H_24_O
Pentalene, Octahydro-1-(2-Octyldecyl)	12.24	0.29	362	C_26_H_50_
1,2-15,16-Diepoxyhexadecane	12.33	0.72	254	C_16_H_30_O_2_
2,5-Furandione, 3-(Dodecenyl) Dihydro	15.05	0.35	266	C_16_H_26_O_3_
14-á-H-Pregna	15.73	0.6	288	C_21_H_36_
12-Methyl-E,E-2,13-octadecadien-1- ol	16.83	1.68	280	C_19_H_36_O

RT: retention time; MW: molecular weight; MF: molecular formula.

**Table 2 biology-09-00352-t002:** Effect of betamethasone and/or argan oil on erythrogram parameters of different treated groups.

Parameters	Experimental Groups					
	Control	BM	AO/0.5	AO/1	BM+AO/0.5	BM+AO/1
RBCs (×10^6^)	8.04 ± 0.12 ^a^	6.81 ± 0.59 ^b^	7.89 ± 0.20 ^a^	7.83 ± 0.31 ^a^	7.26 ± 0.11 ^ab^	7.54 ± 0.14 ^ab^
Hb (g/dl)	15.90 ± 0.05 ^a^	12.88 ± 1.16 ^b^	15.40 ± 0.54 ^a^	15.60 ± 0.21 ^a^	14.72 ± 0.058 ^a^	14.88 ± 0.097 ^a^
PCV (%)	46.00 ± 0.63 ^a^	41.00 ± 2.79 ^b^	45.80 ± 0.58 ^a^	46.00 ± 0.63 ^a^	44.40 ± 0.81 ^ab^	45.00 ± 0.71 ^a^
MCV (fl)	57.29 ± 1.46 ^a^	60.79 ± 1.99 ^a^	58.20 ± 1.64 ^a^	59.06 ± 1.84 ^a^	61.21 ± 1.60 ^a^	60.38 ± 1.66 ^a^
MCH (pg)	19.79 ± 0.32 ^a^	18.89 ± 0.23 ^a^	19.51 ± 0.39 ^a^	20.05 ± 0.84 ^a^	20.29 ± 0.34 ^a^	19.77 ± 0.43 ^a^
MCHC (%)	34.59 ± 0.49 ^a^	31.19 ± 0.99 ^b^	33.63 ± 1.17 ^ab^	33.94 ± 0.71 ^a^	33.20 ± 0.66 ^ab^	33.09 ± 0.52 ^ab^

Values are means ± SE (*n* = 8). Different letters (^a^,^b^) in the same row indicate significant differences at *p* < 0.05. BM: betamethasone, AO: argan oil, RBCs: red blood cell count, Hb: hemoglobin concentration, PCV: packed cell volume, MCV: mean corpuscular volume, MCH: mean corpuscular hemoglobin and MCHC: mean corpuscular hemoglobin concentration.

**Table 3 biology-09-00352-t003:** Effect of betamethasone and/or argan oil on leukogram parameters and blood platelets of different treated groups.

Parameters	Experimental Groups					
	Control	BM	AO/0.5	AO/1	BM + AO/0.5	BM + AO/1
TWBCs (×10^3^)	11.50 ± 0.04 ^b^	12.68 ± 0.08 ^a^	11.54 ± 0.27 ^b^	11.50 ± 0.24 ^b^	11.36 ± 0.12 ^b^	11.44 ± 0.11 ^b^
Neutrophils (×10^3^)	3.13 ± 0.04 ^bc^	4.77 ± 0.11 ^a^	3.12 ± 0.21 ^bc^	2.99 ± 0.11 ^c^	3.48 ± 0.07 ^b^	3.36 ± 0.07 ^b^
Lymphocytes (×10^3^)	7.82 ± 0.050 ^a^	6.97 ± 0.057 ^b^	7.77 ± 0.18 ^a^	7.87 ± 0.19 ^a^	7.18 ± 0.11 ^b^	7.35 ± 0.13 ^b^
Monocytes (×10^3^)	0.53 ± 0.03 ^d^	0.94 ± 0.03 ^a^	0.62 ± 0.03 ^c^	0.62 ± 0.03 ^c^	0.70 ± 0.02 ^bc^	0.73 ± 0.03 ^b^
Eosinophils (×10^3^)	0.02 ± 0.02 ^a^	0.00 ± 0.00 ^a^	0.02 ± 0.02 ^a^	0.02 ± 0.02 ^a^	0.00 ± 0.00 ^a^	0.00 ± 0.00 ^a^
Platelets (×10^3^)	353.0 ± 0.45 ^a^	335.6 ± 0.51^c^	354.0 ± 1.05 ^a^	353.4 ± 0.81 ^a^	349.2 ± 0.97 ^b^	349.4 ± 1.47 ^b^

Values are means ± SE (*n* = 8). Different letters (^a^,^b^,^c^) in the same row indicate significant differences at *p* < 0.05. BM: betamethasone, AO: argan oil, TWBCs: total white blood cells.

**Table 4 biology-09-00352-t004:** The main histopathological changes recorded in kidneys of the different treated groups.

Lesions	Experimental Groups					
	Control	BM	AO/0.5	AO/1	BM+AO/0.5	BM+AO/1
Vascular Changes	−	++++	−	−	+	+
Degenerative Changes	−	+++	−	−	+	+
Inflammatory Changes	−	++	−	−	+	+

The histopathological changes are graded as follows: (−) indicates normal appearance, (+) indicates mild changes, (++) indicates moderate changes, (+++) indicates severe changes, and (++++) indicates very severe changes. BM: betamethasone and AO: argan oil.

**Table 5 biology-09-00352-t005:** Semiquantitative analysis of Bax, Bcl-2, Bax/Bcl-2 ratio, caspase-3, and PCNA immune-staining in the kidneys of different treated groups.

IHC	Experimental Groups					
	Control	BM	AO/0.5	AO/1	BM + AO/0.5	BM + AO/1
Bax (% of positive cells/HPF)	9.85 ± 0.61 ^c^	28.24 ± 1.43 ^a^	8.37 ± 0.54 ^c^	7.25 ± 0.31 ^c^	15.94 ± 1.01 ^b^	14.21 ± 1.05 ^b^
Bcl-2 (% of positive cells/HPF)	27.33 ± 0.88 ^a^	5.36 ± 0.17 ^b^	24.35 ± 0.54 ^a^	25.40 ± 0.44 ^a^	28.67 ± 0.50 ^a^	28.82 ± 0.13 ^a^
Bax/Bcl-2 Ratio (% of positive cells/HPF)	0.36 ± 0.06 ^b^	5.27 ± 0.73 ^a^	0.34 ± 0.02 ^b^	0.29 ± 0.03 ^b^	0.56 ± 0.03 ^b^	0.49 ± 0.02 ^b^
Caspase-3 (% of positive cells/HPF)	7.07 ± 0.72 ^c^	39.87 ± 2.00 ^a^	6.42 ± 0.36 ^c^	5.49 ± 0.64 ^c^	18.48 ± 1.73 ^b^	15.75 ± 1.92 ^b^
PCNA (% of positive cells/HPF)	32.75 ± 0.39 ^a^	10.67 ± 0.36 ^b^	33.35 ± 0.49 ^a^	34.13 ± 0.46 ^a^	30.48 ± 0.43 ^a^	31.70 ± 0.31 ^a^

The values are expressed as means ± SE of immunoreactive renal tubular cells; different letters in the same row indicate significant differences at *p* < 0.05. BM: betamethasone, AO: argan oil. IHC: immunohistochemistry, Bcl-2: B-cell lymphoma 2, Bax: Bcl-2-associated X, PCNA: proliferating cell nuclear antigen.

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
