# Peer review of "The Antioxidant, Anti-Apoptotic, and Proliferative Potency of Argan Oil against Betamethasone-Induced Oxidative Renal Damage in Rats"

_biology, 2020, doi:10.3390/biology9110352_

Round 1

Reviewer 1 Report

The authors resolved my questions satisfactorily and the paper is improved in this revised version

Reviewer 2 Report

No further comments

This manuscript is a resubmission of an earlier submission. The following is a list of the peer review reports and author responses from that submission.

Round 1

Reviewer 1 Report

In the present study, Orabi and colleagues aimed to investigate the beneficial effects of Argan oil (AO) against the nephrotoxicity caused by high doses and/or prolonged administration of betamethasone (BM). By using a biochemical approach, the Authors suggested that AO can induce an improvement in the haematological status and in renal oxidant/antioxidant balance, as well as in renal function. Moreover, the Authors indicated that AO was able to mitigate the typical histopathological alterations BM-induced by targeting B-cell lymphoma 2 (Bcl-2) and proliferating cell nuclear antigen (PCNA).

The study is of patho-physiological interest, the methodological approach and experimental design appear appropriate. However, in my opinion, important points need to be resolved before considering the paper suitable for publication.

Criticism are indicated below:

Points:

  • An important limitation regards the apoptosis involvement:
  • The Authors claim that a significant reduction in apoptosis has been observed in BM+AO experimental groups. However, the evaluation of Bcl-2 alone is not sufficient to conclude that a reduction in apoptosis occurs. In this regard, the authors should evaluate other pro-apoptotic/anti-apoptotic markers and refer to the Bax/Bcl-2 ratio. They could also use molecular approaches to evaluate apoptotic markers.
  • in Discussion section, the Authors refer to mitochondria-mediated apoptosis and the activation of caspase-3; however, no data relating the effector caspase is present in the results, and no experiments on isolated mitochondria have been performed in the study. Authors should integrate this issue with the point 1.
  • Again, the authors report “These findings suggest that these regulators of apoptosis play crucial roles in the repairing process of the damaged renal tubules…”. However, the regulatory role of apoptosis in this experimental model should be further demonstrated. The authors should remodulated this part and integrate it with additional findings.
  • An important point is related to the blood pressure. Did the authors measured the blood pressure in rats exposed to the treatments? How AO could affect this aspect?
  • The authors should discuss what could be the mechanism by which AO ameliorates the hematological alteration induced by BM. Do they have idea on which molecular axis can be activated by AO.
  • In Materials and Methods Overall, the authors should be reserved more attention to this section:
  • the experimental schedule (i.e. dose of AO, the treatment times, etc...) should be supported by specific references. In particular, the experimental trials as rational to justify the specific doses of AO appear inadequate; did the authors establish this empirically or they have some unpublished/preliminary evidences? In addition, do they have idea on the pharmacological toxicity profile of AO?
  • the “Chemicals” section needs to be clarified. The authors should include the specific identificative codes for the reagents used, that should be indicated for experimental purpose only. Also, the commercial kits used should be specified;
  • the anaesthetic used and the percentage of isoflurane should be specified;
  • there is no paragraph relating the sacrifice of animals and the preparation of homogenates for tissue biochemical analyses. The Authors should add these information;
  • Statistical representation should be revised. The statistical differences on figures appear difficult for readers to interpret;
  • I suggest to not use acronyms in Abstract, that should be also improved and better harmonized
  • Several typos are present in the text.

Author Response

Dear Reviewer,

I really appreciate your valuable comments that will improve our manuscript. The required changes have been highlighted with red color and a cover letter has been done to explain point-by-point the details of the revisions in the manuscript and our responses to the comments.

Reviewer: 1#

In the present study, Orabi and colleagues aimed to investigate the beneficial effects of Argan oil (AO) against the nephrotoxicity caused by high doses and/or prolonged administration of betamethasone (BM). By using a biochemical approach, the Authors suggested that AO can induce an improvement in the haematological status and in renal oxidant/antioxidant balance, as well as in renal function. Moreover, the Authors indicated that AO was able to mitigate the typical histopathological alterations BM-induced by targeting B-cell lymphoma 2 (Bcl-2) and proliferating cell nuclear antigen (PCNA).

The study is of patho-physiological interest, the methodological approach and experimental design appear appropriate. However, in my opinion, important points need to be resolved before considering the paper suitable for publication.

1)

The Authors claim that a significant reduction in apoptosis has been observed in BM+AO experimental groups. However, the evaluation of Bcl-2 alone is not sufficient to conclude that a reduction in apoptosis occurs. In this regard, the authors should evaluate other pro-apoptotic/anti-apoptotic markers and refer to the Bax/Bcl-2 ratio. They could also use molecular approaches to evaluate apoptotic markers.

§  The immunohistochemistry investigation of Bax has been evaluated and the obtained results and their interpretations have been included all over the manuscript. Also Bax/Bcl-2 ratio was included,

2)

In Discussion section, the Authors refer to mitochondria-mediated apoptosis and the activation of caspase-3; however, no data relating the effector caspase is present in the results, and no experiments on isolated mitochondria have been performed in the study. Authors should integrate this issue with the point 1.

Also, The immunohistochemistry investigation of caspase-3 has been evaluated and the obtained results and their interpretations have been included all over the manuscript and integrated with the first point.

3)

Again, the authors report “These findings suggest that these regulators of apoptosis play crucial roles in the repairing process of the damaged renal tubules…”. However, the regulatory role of apoptosis in this experimental model should be further demonstrated. The authors should remodulated this part and integrate it with additional findings.

§   This part was remodulated and integrated with the additional new obtained findings of Bax and caspase-3 investigations.

4)

An important point is related to the blood pressure. Did the authors measured the blood pressure in rats exposed to the treatments? How AO could affect this aspect?

  • We didn't measure the blood pressure in rats exposed to the different treatments and we didn't include this point in the manuscript. However, previous literatures study the effect of AO on blood pressure and confirmed that AO administration reduced the mean blood pressure in spontaneously hypertensive rats (Berrougui et al., 2004). Also, AO treatment reduced elevation of blood pressure, hyperglycemia, and insulin resistance through its antioxidative properties, mainly via decreasing NADPH oxidase activity and production of basal superoxide anion in aorta (El Midaoui et al., 2016). Also, Driss et al. (2004) reported that AO was found to increase the molar ratio of a-tocopherol to total cholesterol and a-tocopherol concentration.

5)

The authors should discuss what could be the mechanism by which AO ameliorates the hematological alteration induced by BM. Do they have idea on which molecular axis can be activated by AO.

Argan oil could ameliorate the hematological alterations induced by BM. The improved hematological parameters may be attributed to the recorded improvement in kidney functions following AO treatment due to the essential role of kidney in the production of hematopoietic growth factors erythropoietin and thrombopoietin hormones which regulate the erythropoiesis and thrombopoiesis processes [54]. Moreover, AO could ameliorate the hematological alterations induced by BM, mainly through its antioxidant and anti-inflammatory activities [18]. (line 458-463)

6)

The experimental schedule (i.e. dose of AO, the treatment times, etc...) should be supported by specific references. In particular, the experimental trials as rational to justify the specific doses of AO appear inadequate; did the authors establish this empirically or they have some unpublished/preliminary evidences? In addition, do they have idea on the pharmacological toxicity profile of AO?

The selected doses were based on previous literatures that used AO at doses ranged from 2-10 ml/kg b.wt, however this study used 2 different doses as experimental trials to evaluate the effectiveness of AO at lower doses. Regarding its pharmacological toxicity profile, AO has been used as a food and as a food ingredient, and has been applied to the skin for centuries, therefore its acute and chronic toxicity is assumed to be nil, particularly when orally administered at ordinary doses (El Monfalouti et al., 2010; Guillaume and Charrouf, 2011).

7)

The “Chemicals” section needs to be clarified. The authors should include the specific identificative codes for the reagents used, that should be indicated for experimental purpose only. Also, the commercial kits used should be specified.

§  The catalogue numbers were included and the commercial kits were specified. (Line150-152 and 154-156).

8)

The anaesthetic used and the percentage of isoflurane should be specified.

§  Rats were anaesthetized by inhalation of isoflurane as the following:

 Animals were exposed to a piece of cotton soaked in isoflurane underneath glass funnel. Once the animal has reached the desired depth of anesthesia, animals were removed quickly and carefully and were placed on a clean work surface for blood sample collection.

9)

There is no paragraph relating the sacrifice of animals and the preparation of homogenates for tissue biochemical analyses. The Authors should add these information

§  Sacrificing of animals (Line129-130) and the preparation of tissue homogenates (Line133-139) were included in details in the manuscript.

10)

Statistical representation should be revised. The statistical differences on figures appear difficult for readers to interpret.

§  The statistical representations depend on the mentioned different superscripts “a, b or c” in the figures indicate significant differences at PË‚0.05 (specified in the legend of figures). (Line 246 & 259).

(i.e, chart that contains a superscript "a" is statistically different from chart with "b" or "c" superscripts; however, insignificantly different from that contains "a" or "ab" or "ac" superscript).

11)

I suggest to not use acronyms in Abstract, that should be also improved and better harmonized

§  The acronyms were deleted and the abstract was amended to include the new findings and remodulated to match the journal requirements (about 200 words maximum).

12)

Several typos are present in the text.

  •  

The manuscript was revised precisely for any editing and formatting errors.

Reviewer 2 Report

The authors show the potential therapeutic activity of the argan oil in curbing the effect of betamethasone overdose in rat kidney. Although the argan oil seems to ameliorate some of the features of BM overdose, the authors overstate those effects. Quite poor is also the attempt to explain the AO mechanism although little improvement can be done, for example:

  • very nice it is the histochemical downregulation of Bcl-2 in BM and the consequent improvement due to AO co-administration. What I would suggest is to stain some slides also for Bax, since Bcl-2 prevents Bax from perforating the outer mitochondrial membrane, causing cytochrome c to leak out and apoptosis.
  • It is well known that corticosteroids have huge impact on liver function. Some H&E staining of the liver of all the groups would add a great improvement to the study.
  • It would make more attractive and informative the study the addition of some in vitro experiments, to see if the AO exerts beneficial activity to a specific cell type.

  • line 54: add “s” to hormone or please specify which hormone BM has effect on.

  • Line 63: “Thus, natural products rich in antioxidants 63 are needed to ameliorate such adverse effects.”. It is a quite bald statement. Please expand a little more why natural products rich in antioxidants ameliorate the adverse effects.

  • Line 70: please could you explain how the rectally application of AO in rats with colorectal anastomosis support the statement of the use of AO in shampoos and moistures?

  • Line 81: the rats in the group of BM received 1 mg/kg of treatment for three weeks. However, it is not clear how the calculation was made since the commercial available contain: “Each ml of  DIPROFOS  suspension contains betamethasone dipropionate equivalent to 5 mg betamethasone and betamethasone sodium  phosphate  equivalent  to  2 mg  betamethasone in a sterile buffered and preserved vehicle” according to datasheet.

  • Line 115: the last two group are not in bold. Also, it is not clear if BM and AO were administered orally or BM intramuscular and AO orally. Please specify.

  • Line 120: please specify the volume of the blood taken and how the chemical assessment for Hb (g/dL), MCV (fl) and MCH (pg) is performed.

  • Line 135: if commercial kits have been used, please add company, and catalogue number and remove the references.

  • Line 138: same as line 135.

  • Line 147: please summarize the parameters used for scoring even though the reference given did not include any kidney/renal-damage part. If it does, please specify it.

  • Line 152: please specify the diluent for H2O2 (also fix the numbers as subscripts).

  • Line 162: the authors state that “Zero indicates negative staining; 1 indicates Ë‚25% of positive cells/ field; 2 indicates 25-50%; 3 162 indicates 51-75%; and 4 indicates ˃75%.”. However, no such scoring (AI%) is present throughout the paper. Specifically, in table 5 is reported, I think, the AI% = (number of positive 160 cells/total number of calculated cells) × 100 without any score from 0 to 4. Please explain.

  • Line 176: in 3.1 section, the authors made a list of all the compounds detected by GC. However, the same compounds are listed in Table 1. Please delete the table or the 3.1 section unless the authors want add a comment on the two most abundant compound found: “Cyclopentasiloxane, decamethyl-“ and “Phenol, 2,6-BIS(1,1-Dimethylethyl)-4 -Methyl”. Also format the content of the compound name column on the left. It is difficult to read the name of the compounds.

  • Table 2: please specify in the legend what the letters “a and b” means, but above all which groups are statistically compared with. Also specify the “N” numbers.

  • Table 3: same as table 2.

  • Figure 1: please specify in the legend what the letters “a and b and c” means and add “N” numbers. The high level of urea could indicate in some instances a liver damage which could affect the kidney as reported

  • Line 242: according to the figure 2, apart from MDA, it does not seem a “marked improvement” of the co-treatment for NO and GSH. If not so, please delete the word “marked”.

  • Line 260: Although it is clear from the table 4 that groups AO/0.5 and AO/1 have similar “score” to control group, in Figure 3, the same conditions present some congestion (yellow arrow). On the same line, the histology of the groups BM+AO/0.5 and BM+AO/1 (Figure 3 E and F, respectively) look more like the BM group than Control group especially for vascular and inflammatory damage. Please, choose different pictures or restate the lines from 260 to 263.

Author Response

Dear Reviewer,

I really appreciate your valuable comments that will improve our manuscript. The required changes have been highlighted with red color and a cover letter has been done to explain point-by-point the details of the revisions in the manuscript and our responses to the comments.

Reviewer: 2#

The authors show the potential therapeutic activity of the argan oil in curbing the effect of betamethasone overdose in rat kidney. Although the argan oil seems to ameliorate some of the features of BM overdose, the authors overstate those effects. Quite poor is also the attempt to explain the AO mechanism although little improvement can be done, for example:

1)

Very nice it is the histochemical downregulation of Bcl-2 in BM and the consequent improvement due to AO co-administration. What I would suggest is to stain some slides also for Bax, since Bcl-2 prevents Bax from perforating the outer mitochondrial membrane, causing cytochrome c to leak out and apoptosis.

§  The immunohistochemistry investigation of Bax has been evaluated and the obtained results and their interpretations have been included all over the manuscript.

2)

It is well known that corticosteroids have huge impact on liver function. Some H&E staining of the liver of all the groups would add a great improvement to the study.

§  Regarding to our findings, BM induced various histopathological changes in liver tissue (unpublished data). However, this study aimed mainly to evaluate the effect of AO on BM- induced renal damage.

3)

It would make more attractive and informative the study the addition of some in vitro experiments, to see if the AO exerts beneficial activity to a specific cell type.

We didn't perform in vitro studies based on the previous literatures that studied and approved the beneficial activities of AO:

§  The preventive and therapeutic effects of AO in lymphoblastic leukemia cell lines (Aribi et al., 2016).

§  The antioxidant and cytoprotective effects of AO in 158N murine oligodendrocytes cultured with 7-Ketocholesterol (Badreddine 2017).

§  The anti- inflammatory, anti-proliferative and anti-bacterial activities of AO (Lall, et al., 2019).

4)

Line 54: add “s” to hormone or please specify which hormone BM has effect on.

§  We added “s” to hormone to be hormons (Line 54).

5)

Line 63: “Thus, natural products rich in antioxidants are needed to ameliorate such adverse effects”. It is a quite bald statement. Please expand a little more why natural products rich in antioxidants ameliorate the adverse effects.

§  The paragraph was revised, expanded and remodulated in the introduction section of the manuscript "Thus, natural products rich in antioxidants are needed for curbing such adverse effects, mainly via its remarkable ability to scavenge ROS, modulate antioxidant/pro-oxidant enzymes and/or transcription factors, enhance cellular antioxidant enzymes, and formation of bioactive metabolites [14]". (Line 63-66)

6)

Line 70: please could you explain how the rectally application of AO in rats with colorectal anastomosis support the statement of the use of AO in shampoos and moistures?

§   

This statement was already reported in this literature (Barlas et al., 2018) in the introduction section without referring to a certain reference. However, we found another literature recorded the same statement with referring to a certain reference (Charrouf and Guillaume, 2011). So, this reference was replaced within the introduction and reference sections. (Line 72)

The corrected reference is "Charrouf Z, Guillaume D. Argan oil and other argan products; use in dermocosmetology. Eur J Lipid Sci Technol2011;113:403-408".

7)

Line 81: the rats in the group of BM received 1 mg/kg of treatment for three weeks. However, it is not clear how the calculation was made since the commercial available contain: “Each ml of DIPROFOS suspension contains betamethasone dipropionate equivalent to 5 mg betamethasone and betamethasone sodium phosphate equivalent to 2 mg betamethasone in a sterile buffered and preserved vehicle” according to datasheet.

§  Final concentration of betamethasone is (2mg+5mg/1 ml), it means 7 mg/ml {Diprfos}.

§  7 mg/1 ml = 7000 microg/1000 microL

§  We take1 ml from betamethasone ampoule to 9 ml sterile water for injection®.

§  The final concentration in 10 ml was 0.7mg/ml (0.7 mg/1000 microL.).

§  Every 1000 gm bw of rats required 1 mg BM: (1 mg/kg) as a dose rate.

§  Every 100 gm bw of rats required 0.1 mg BM and its equivalent in µL was 142.85 µL.

§  So, the dose was 142.85 µL/0.1 mg/100 gm bw.

§  And 285.71 µL/0.2 mg/200 gm bwt rat. And so on,…………

8)

Line 115: The last two groups are not in bold. Also, it is not clear if BM and AO were administered orally or BM intramuscular and AO orally. Please specify.

§  The last two groups were rewritten and amended according to the reviewer comment (line 121 & 123).

9)

Line 120: please specify the volume of the blood taken and how the chemical assessment for Hb (g/dL), MCV (fl) and MCH (pg) is performed.

§  The volume of blood taken from each rat was about 3ml, about 0.5 ml (500ul) for hemogram evaluation and 2.5 ml (1500ul) placed in a plain tube for separating serum in order to evaluate serum kidney biomarkers.

§  The erythrogram (RBCS, PCV, Hb, MCV, MCH and MCHC), leukogram, and blood platelets were assessed using BeneSphera™ Brand3 Part differential veterinary hematology analyzer H32 (Avantor Performance Materials, Inc, Netherlands, Model: H32vet. Serial No: 931716004) in the Central Lab of Chemical Diagnosis and Blood Research, accredited lab according to ISO17025/2017, Faculty of veterinary medicine, University of Sadat City. (Line 146-148)

10)

Line 135: if commercial kits have been used, please add company and catalogue number and remove the references.

§  Company, and catalogue number were added and the references were deleted (Line150-152).

11)

Line 138: same as line 135.

§  Company, and catalogue number were added and the references were deleted (Line154-156).

12)

Line 147: please summarize the parameters used for scoring even though the reference given did not include any kidney/renal-damage part. If it does, please specify it.

§  The parameters used for scoring of renal lesions with specifying a reference were added to the methodology section (Line 161-167) and also, the scoring was already indicated in "table 4 legends".

13)

Line 152: please specify the diluent for H2O2 (also fix the numbers as subscripts).

  •  

 The diluent for H2O2 is water, and the numbers of H2O2 were fixed as subscripts (Line 173).

14)

Line 162: the authors state that “Zero indicates negative staining; 1 indicates Ë‚25% of positive cells/ field; 2 indicates 25-50%; 3 indicates 51-75%; and 4 indicates ˃75%”. However, no such scoring (AI%) is present throughout the paper. Specifically, in table 5 is reported, I think, the AI% = (number of positive 160 cells/total number of calculated cells) × 100 without any score from 0 to 4. Please explain.

·         The lesions were scored according to AI%, and the statement of scoring from 0-4 was deleted in the revised manuscript as it was an error typing. (Line 183-188)

15)

Line 176: in 3.1 section, the authors made a list of all the compounds detected by GC. However, the same compounds are listed in Table 1. Please delete the table or the 3.1 section unless the authors want add a comment on the two most abundant compound found: “Cyclopentasiloxane, decamethyl-“ and “Phenol, 2,6-BIS(1,1-Dimethylethyl)-4 -Methyl”. Also format the content of the compound name column on the left. It is difficult to read the name of the compounds.

§  3.1 section was amended according to the reviewer comment (Line 197-199).

Concerning the GC/MS findings, the two most abundant compound found in AO are Cyclopentasiloxane, decamethyl- and “Phenol, 2,6-BIS(1,1-Dimethylethyl)-4 –Methyl. Cyclopentasiloxane, decamethyl is classified as a cyclomethicone thatis commonly used in cosmetics, such as deodorantssunblockhair sprays and skin care products. It is also used as part of silicone-based personal lubricants [52], while Phenol, 2,6-bis(1,1-dimethylethyl)-4-methyl is the phenolic antioxidants also known as 2,6-ditertiary-butyl paracresol (DBPC) or  butylatedhydroxytoluene (BHT)  [53]. (Line 447-453)

16)

Table 2: please specify in the legend what the letters “a and b” means, but above all which groups are statistically compared with. Also specify the “N” numbers.

§  Different letters “a and b” in the same row indicate significant differences at PË‚0.05 (already was specified in the legend) and numbers of rats/ group were added (n=8). (Line 216)

(i.e, row that contains a superscript "a" is statistically different from raw with "b" or "c" superscripts; however, insignificantly different from that contains "a" or "ab" or "ac" superscript).

17)

Table 3: same as table 2.

§  Different letters “a, b and c” in the same row indicate significant differences at PË‚0.05 (already was specified in the legend) and numbers of rats/ group were added (n=8). (Line 233)

18)

Figure 1: please specify in the legend what the letters “a and b and c” means and add “N” numbers. The high level of urea could indicate in some instances a liver damage which could affect the kidney as reported

§  Different superscripts “a, b or c” indicate significant differences at PË‚0.05 (specified in the legend) and numbers of rats/ group were added (n=8) (Line 246).

(i.e, chart that contains a superscript "a" is statistically different from charts with "b" or "c" superscripts; however, insignificantly different from that contains "a" or "ab" or "ac" superscript).

§  Regarding to our findings, BM induced alterations in liver biomarkers (unpublished data). However, this study aimed mainly to evaluate the effect of AO on BM- induced renal damage.

19)

Line 242: according to the figure 2, apart from MDA, it does not seem a “marked improvement” of the co-treatment for NO and GSH. If not so, please delete the word “marked”.

The word “marked” was deleted. (Line 254)

20)

Line 260: Although it is clear from the table 4 that groups AO/0.5 and AO/1 have similar “score” to control group, in Figure 3, the same conditions present some congestion (yellow arrow). On the same line, the histology of the groups BM+AO/0.5 and BM+AO/1 (Figure 3 E and F, respectively) look more like the BM group than Control group especially for vascular and inflammatory damage. Please, choose different pictures or restate the lines from 260 to 263.

§  The figures were replaced according to the reviewer comment and the comment within the text and the legends were also revised.
